# The Tug-of-War between Plants and Viruses: Great Progress and Many Remaining Questions

**DOI:** 10.3390/v11030203

**Published:** 2019-02-28

**Authors:** Xiaoyun Wu, Adrian Valli, Juan Antonio García, Xueping Zhou, Xiaofei Cheng

**Affiliations:** 1College of Agriculture, Northeast Agricultural University, Harbin 150030, China; wxy5551@126.com; 2Centro Nacional de Biotecnología (CNB-CSIC), Campus Universidad Autónoma de Madrid, 28049 Madrid, Spain; avalli@cnb.csic.es (A.V.); jagarcia@cnb.csic.es (J.A.G.); 3State Key Laboratory of Rice Biology, Institute of Biotechnology, Zhejiang University, Hangzhou 310058, China; zzhou@zju.edu.cn; 4State Key Laboratory for Biology of Plant Diseases and Insect Pests, Institute of Plant Protection, Chinese Academy of Agricultural Sciences, Beijing 100193, China

**Keywords:** innate immunity, RNA silencing, translation repression, autophagy, dominant resistance

## Abstract

Plants are persistently challenged by various phytopathogens. To protect themselves, plants have evolved multilayered surveillance against all pathogens. For intracellular parasitic viruses, plants have developed innate immunity, RNA silencing, translation repression, ubiquitination-mediated and autophagy-mediated protein degradation, and other dominant resistance gene-mediated defenses. Plant viruses have also acquired diverse strategies to suppress and even exploit host defense machinery to ensure their survival. A better understanding of the defense and counter-defense between plants and viruses will obviously benefit from the development of efficient and broad-spectrum virus resistance for sustainable agriculture. In this review, we summarize the cutting edge of knowledge concerning the defense and counter-defense between plants and viruses, and highlight the unexploited areas that are especially worth investigating in the near future.

## 1. Introduction

Plant viruses are obligate intracellular parasites that rely almost exclusively on the host cell to accomplish their infection cycle, e.g., genomic information expression, genome replication, and intercellular movement. During the infection, a comprehensive and competitive protein–protein and protein–nucleic acid interaction network is formed between the plant and virus. In general, the interaction network includes mechanisms by which the plant forms antiviral responses and the virus hijacks host factors for proliferation and to cope with plant antiviral defenses. The elaborate balance between the mechanisms used by the plant and the virus determines the outcome of the interaction, whether a virus is pathogenic or not. Understanding the plant–virus interaction network is one of the key goals of virology research. Impressive progress has been made in the knowledge of plant–virus interplay in the past decade, which not only quickly filled in the gap between plant innate immunity against virus and other pathogens, but also revealed unique plant antiviral machinery, such as NUCLEAR SHUTTLE PROTEIN (NSP) INTERACTING KINASE 1 (NIK1)-mediated translation repression and atypical dominant resistance by inhibiting viral protein function. In this review, we present the cutting edge of knowledge concerning the defense–counter-defense between plants and viruses, and outline the areas that may be particularly fruitful for study in the coming years. Due to space limitations, how plant viruses hijack the host’s metabolism for multiplication and dissemination is not included in the review, and readers are referred to the recent reviews [1,2]. Recessive resistance as specific antiviral resistance machinery due to the loss of key host factors for virus proliferation is also not discussed here; instead, the readers are referred to reviews [3,4,5].

## 2. Plant Antiviral Defenses

Plants are persistently challenged by varied pathogens, including viruses, bacteria, fungi, nematodes, and insects. Unlike the animal counterparts that possess specialized defense cells, plants rely on the capacity of every cell to perceive and defend against the “challengers”. During the millions of years of co-evolution with pathogens, plants have evolved multilayered surveillance mechanisms against pathogens that invade them [6,7]. Innate immunity, RNA silencing, translational repression, and ubiquitination-mediated and autophagy-mediated protein degradation are the major defense mechanisms against viruses in plants [8]. Exploration of the dominant resistance genes in the last decade also resulted in the identification of some antiviral proteins that restrict virus proliferation by directly interacting with and inhibiting viral protein functions (Figure 1).

### 2.1. Innate Antiviral Immunity

According to current plant immunity descriptions, there are two layers of plant immune responses against microbial pathogens. First, plant perception of the microorganism at the cell surface through the recognition of certain conserved pathogen-associated molecular patterns (PAMPs) by the extracellular surface pattern recognition receptor (PRR) initiates the so-called PAMP-triggered immunity (PTI) [9,10]. Plant PRRs fall into two major classes: the plasma membrane-localized receptor kinases (RKs) and the receptor-like proteins (RLPs) [11,12]. After the successful perception of PAMPs, PRRs immediately dimerize and associate with cofactors, such as SOMATIC EMBRYOGENESIS RECEPTOR-LIKE KINASE (SERKs) and SUPPRESSOR OF BIR1-1 (SOBIR1), and trigger a series of downstream intracellular signaling events including, but not limited to, oxidative burst, ion influx, the increased biosynthesis of defense hormones, and the activation of MITOGEN-ACTIVATED PROTEIN KINASES (MAPKs) that finally lead to resistance responses, e.g., the expression of pathogenesis-related (*PR*) genes, the synthesis and deposition of callose at the plasmodesmata (PD), and strengthening of the cell wall [13,14]. Occasionally, the activation of PTI also results in the hypersensitive response (HR), which is a specific type of programmed cell death (PCD) that occurs in the cells at the site of infection and causes discrete necrotic spots [15]. At present, a large number of plant PRRs and their respective PAMPs have been identified and characterized [12].

Despite the mass of knowledge on plant PTI against other phytopathogens, little is known about plant PTI against plant viruses, as they are historically viewed as non-PAMP coding pathogens [7]. Nevertheless, several lines of evidence suggest that PTI also plays a pivotal role in both incompatible and compatible plant–virus interaction. First, the Arabidopsis mutants in the PRR coreceptor kinases SERK3 (also known as BRASSINOSTEROID INSENSITIVE1 (BRI1)-ASSOCIATED RECEPTOR KINASE1, BAK1) and/or SERK4 (also called BAK1-LIKE, BKK1) exhibit increased susceptibility to different RNA viruses [16,17]. Similarly, knockdown of the expression of CaLecRK-S.5, which is a transmembrane RK possessing an L-type lectin domain at the N-terminus from pepper (*Capsicum annuum* L.), results in the enhanced susceptibility of pepper to several unrelated pathogens, including two plant viruses, tobacco mosaic virus (TMV, a tobamovirus), and pepper mild mottle virus (PMMoV, a tobamovirus), one bacterium (*Xanthomonas campestris* pv. vesicatoria), and one oomycete pathogen (*Phytophthora capsici*) [18]. Moreover, pre-treatment with β-Aminobutyric acid (BABA), which is a well-known immunity-priming chemical trigger [19], restores the disease resistance in CaLecRK-S.5-silenced plants [18]. Second, the exogenous application of double-strand RNA (dsRNA), which is a well-characterized PAMP in animal antiviral immunity [20,21], also triggers typical PTI responses in Arabidopsis that are dependent on the co-receptor SERK1, but independent of the RNA silencing pathway [22]. Third, the coat protein (CP) of TMV and potato virus X (PVX, a potexvirus) can trigger PTI-like responses in tobacco and Arabidopsis, respectively [23,24]. CP is the outermost component of all the non-enveloped viruses that should have contact with the plant cell surface during the invasion. Thus, it is possible that the PTI-like responses are triggered by unknown PAMPs derived from TMV and PVX CP proteins. Agreeing with the hypothesis, conserved local or overall three-dimensional folding was found in the CPs of viruses with similar particles, such as flexuous filamentous rods [25]. Nevertheless, viral PAMPs, the plant PRRs that perceive them, and cellular signaling components are still elusive at present.

Adapted pathogens can deliver specific proteins (effectors) into plant cells to compromise PTI defenses. To further counteract the effectors, plants have evolved an additional monitoring system that relies on particular intracellular receptors known as R proteins to directly or indirectly recognize the microbial effectors that are used to subvert PTI and trigger the so-called effector-triggered immunity (ETI) [26]. Over the past decade, a large number of *R* genes that mediate resistance against various plant viruses have been cloned, as they have apparent practical application value [27,28]. Functional R proteins typically contain an N-terminal Toll/Interleukin-1 receptor homology (TIR) domain or a coiled-coil (CC) domain, a central nucleotide-binding (NB) domain, and a C-terminal leucine-rich repeat (LRR) domain [6]. In addition, non-canonical domains have also been identified in some plant R proteins [29]. The NB-LRR domain plays a pivotal role in pathogen perception, although the non-canonical domain might also have a role in enhancing the recognition [30]. Similar to PTI, the successful perception of an effector by an R protein will trigger a series of downstream signaling events that lead to resistance responses. The direct consequence of defense in most R-mediated resistance responses is HR [31]. However, some *R* genes, such as the potato *Rx1* gene, confer an extreme immunity that is not associated with HR [32]. Rx1 recognizes the PVX CP by the leucine-rich repeat domain, and it is then translocated to the nucleus by the RAN GTPase-ACTIVATING PROTEIN 2 RanGAP2 to activate the resistance against PVX, possibly via a Golden2-like transcription factor (NbGlk1) [33,34,35]. Thus, it was proposed that resistance and HR are separate responses in R-mediated resistance [36]. However, HR has an apparent function in restricting viral proliferation, as inhibiting or delaying the cell death via high-temperature treatment or the transgenic over-expression of HR suppressors, e.g., the baculovirus p35 protein and the endogenous plastid-localized protein DS9, results in the systemic infection of TMV on tobacco plants carrying the resistance *N* gene [37,38,39].

The components that are required for R-mediated resistance against virus infection largely overlap with those for other phytopathogens, suggesting the convergence of plant ETI [7]. For instance, SUPPRESSOR OF THE G2 ALLELE OF SKP1 (SGT1), REQUIRED FOR MLA12 RESISTANCE1 (RAR1), and HEAT SHOCK PROTEIN90 (HSP90) form the SGT1/RAR1/HSP90 complex that is required for both the *N*-mediated and *Rx*-mediated resistance against TMV and PVX, respectively, and immune responses against bacterial infections [40,41,42,43]. The complex formed by ENHANCED DISEASE SUSCEPTIBILITY1 (EDS1), PHYTOALEXIN DEFICIENT4 (PAD4), and SENESCENCE-ASSOCIATED GENE101 (SAG101) regulates HR TO TCV INFECTION (HRT)-mediated resistance against turnip crinkle virus (TCV), bacteria, fungi, and even cold tolerance [44,45,46,47,48].

Both PTI and ETI can result in the establishment of resistance in the distal, non-infected tissues, which is a phenomenon called systemic acquired resistance (SAR) [49]. Intriguingly, the resistance is not only limited to the previous pathogens, but also to other unrelated pathogens, e.g., viruses, bacteria, oomycetes, and fungi [50]. Moreover, SAR can even be passed onto progeny through epigenetic regulation [51]. Salicylic acid (SA) is believed to be the key plant hormone for establishing SAR [52]. NON-EXPRESSOR OF PR1 (NPR1), an ankyrin domain-containing protein, is the core component responding to SA signaling and establishing SAR [49,53]. Besides NPR1, the model plant Arabidopsis encodes other two NPR1-like homologs, namely, NPR3 and NPR4 [54]. It was proposed that the local innate immune responses cause increments of SA concentration in the non-infected tissues that disrupt NPR1–NPR3, but promote interaction between NPR1–NPR4, and finally induce the expression of PATHOGEN-RELATED (*PR*) genes [55]. However, this hypothesis is challenged by a recent study, which provides genetic evidence supporting NPR3 and NPR4 as redundant immune repressors [56]. Nevertheless, NPR1 is not only required for SAR, but is also necessary for robust immune responses in both PTI and ETI, as the knockout of *NPR1* attenuates *PR* gene expression [53]. Thus, NPR1 might be in the core position of plant innate immunity that allows plants to efficiently and dynamically balance immune responses and normal development. Indeed, NPR1 is heavily modified at the post-translational level to ensure the proper regulation of the immune responses [53].

### 2.2. RNA Silencing

RNA silencing, which is also called RNA interference (RNAi), is an evolutionary conserved and sequence-specific mechanism for regulating endogenous gene expression and fighting against foreign nucleic acids such as transposable elements and viruses [57]. The virus-derived double-stranded RNA (dsRNA), which is the key trigger of antiviral RNA silencing, is recognized and processed by plant type III endoribonucleases, the DICER-LIKE (DCL) proteins, into small 20 to 24-nucleotide (nt) RNA duplexes termed virus-derived short-interfering small RNAs (vsiRNAs) [58,59]. The vsiRNAs are incorporated into ARGONAUTE (AGO) proteins that form the core component of the RNA-induced silencing complex (RISC), which is able to directly cleave homologous viral RNAs and/or suppress viral protein translation [60,61,62]. RNA silencing has been recognized as one of the primary antiviral defense mechanisms in the compatible plant–virus interactions, as knockout of the viral suppressor of RNA silencing (VSR) usually results in the loss-of-infectivity of the virus. For instance, a turnip mosaic virus (TuMV, a potyvirus) mutant lacking the major VSR (HcPro) cannot infect wild-type Arabidopsis; however, it regains infectivity on a mutant plant in which the RNA silencing pathway is also impaired [63]. Besides, the phenomenon of symptom recovery is closely associated with enhanced RNA silencing and the dysfunction of VSR activity [64,65]. In fact, it is believed that almost all plant viruses have evolved one or more proteins that are able to interrupt RNA silencing [66]. RNA silencing can also determine the incompatibility of a plant–virus interaction. For instance, Arabidopsis is normally recognized as non-host for PVX. However, PVX can infect Arabidopsis in the presence of pepper ringspot virus (PepRSV, a potyvirus), or with the assistance of the VSR of PepRSV [67]. Further research showed that the inhibition of the replication of PVX in Arabidopsis is mainly dependent on DCL2 and DCL4, and the cooperative activity of AGO2 and AGO5 [67,68,69].

Similar to innate immunity, RNA silencing also (1) perceives extraneous dsRNA molecules in a nucleotide sequence-independent manner; (2) conducts a signal amplification process; and (3) establishes resistance against homologous virus infection on non-infected tissues via the cell-to-cell and systemic movement of vsiRNA duplexes [70,71]. However, unlike PTI, RNA silencing does not initiate typical innate immunity responses, such as *PR* gene expression and HR development, and is independent of small molecules, e.g., SA, reactive oxygen species (ROS), or calcium ions, as secondary messengers for signal amplification.

### 2.3. Translation Repression

The cellular protein biosynthesis is tightly regulated in response to nutrient starvation or stresses. It is believed that GENERAL CONTROL NON-DEREPRESSIBLE-2 (GCN2), which is a serine/threonine-protein kinase, is the key regulator of protein biosynthesis in plants. GCN2 functions by targeting and phosphorylating EUKARYOTIC TRANSLATION INITIATION FACTOR-2A (eIF2α), which binds Met-tRNA and transfers it to the 40S ribosomal subunit to initiate the synthesis of a protein upon bonding to guanosine triphosphate (GTP) [72,73]. The phosphorylation of eIF2α at serine 52 inhibits the recycling of eIF2α-bound GDP to GTP, thereby prohibiting protein synthesis [74]. The activation of GCN2 itself is regulated by an ATP-binding cassette (ABC) family protein, GCN1 [75,76]. The GCN1–GCN2–eIF2α pathway is involved in plant response to many biotic and abiotic stresses, such as wounding and cold stress [76,77]. A direct role for GCN2-mediated protein translation repression in antiviral defense has not been reported, and the infection of wild-type or GCN2-knockout Arabidopsis by turnip yellow mosaic virus (TYMV, a tymovirus) or TCV was shown not to cause the phosphorylation of eIF2α [72]. These data suggest that either GCN2 does not play a role in the plant response to viral infection or, if it does, its role has been co-opted by these two adapted viruses. Further studies are needed in order to fully illustrate these possibilities [78]. Plant viruses thoroughly rely on host protein translation machinery for protein biosynthesis; as a result, loss-of-function of the component(s) of translation machinery that are recruited by plant virus for viral protein synthesis usually begets loss-of-susceptibility to the virus, which is a phenomenon known as recessive resistance [79].

Ribosome-inactivating proteins (RIPs) form a family of proteins that is ubiquitously distributed in plants, mushrooms, and bacteria and can inhibit protein synthesis by depurinating the sarcin/ricin loop (SRL) of rRNA [80,81]. Research demonstrated that many RIPs have potent antiviral, antifungal, and insecticidal activities. The best-characterized RIP with antiviral activity is the pokeweed antiviral protein (PAP) from pokeweed (*Phytolacca americana*). PAP has been demonstrated to be able to reduce the propagation of many plant viruses, e.g., cucumber mosaic virus (CMV, a cucumovirus), PVX, tobacco etch virus (TEV, a potyvirus), alfalfa mosaic virus (AMV, an alfamovirus), potato virus Y (PVY, a potyvirus), brome mosaic virus (BMV, a bromovirus), African cassava mosaic virus (ACMV, a begomovirus), cauliflower mosaic virus (CaMV, a caulimovirus) [82,83,84,85,86], and many animal viruses as well [87]. Moreover, the expression of RIPs in plants is increased under stress conditions, e.g., virus attacking [88], suggesting that they are parts of a plant antiviral mechanism via global translation repression. Indeed, there was a burst of research that sought to utilize RIPs to achieve broad-spectrum antiviral resistance around 1990–2000s; however, their application in agricultural practice is hampered by the potent cytoplasmic toxicity to the plant itself and the animals that consume it [85,89]. How to reduce the cytoplasmic toxicity while maintaining its antiviral activities is a challenge for future research [84].

Recently, another antiviral mechanism that functions by suppression of the global cellular protein synthesis was discovered [90]. NUCLEAR SHUTTLE PROTEIN (NSP) INTERACTING KINASE 1 (NIK1), the key component of this antiviral machinery, was first identified as a geminiviral NSP-interacting host factor [91]. NIK1 is a plasma membrane-anchored LRR protein carrying a functional serine/threonine (Ser/Thr) kinase domain resembling the PRR co-receptor BAK1 [92]. Two additional homologs (NIK2 and NIK3) were identified in Arabidopsis, which together with NIK1 form a separate cluster belonging to group I of the LRR-RK superfamily [93]. Unlike BAK1-mediated resistance, NIK1-mediated antiviral responses do not induce typical innate immune reactions, e.g., the expression of *PR* genes; instead, NIK1 phosphorylates the cytoplasmic RIBOSOMAL PROTEIN 10 (RPL10) and results in the translocation of RPL10 from the cytoplasm to the nucleus [94]. In the nucleus, RPL10 interacts with an myeloblastosis (MYB) domain-containing transcriptional repressor, L10-INTERACTING MYB DOMAIN-CONTAINING PROTEIN (LIMYB), to repress the expression of ribosomal protein genes, and finally leads to the shutdown of global cellular and viral protein synthesis [90].

Besides global translation repression, plants also have developed specific translation repression machinery. One such example is small RNA-related translation repression. As addressed in the above content, the binding of miRNA-associated and short interfering RNA (siRNA)-associated RISCs to cellular or viral RNA, in addition to endonucleolytic cleavage, also causes direct translation inhibition [61,62,95,96]. Similar to RNA silencing, the specificity of small RNA-related translation repression is determined by the sequence of small RNA; thus, the overall protein synthesis is largely not affected [97]. Small RNA-related translation repression can play an important role in plant–virus interaction, and even determines the outcome of the battle. For instance, symptom recovery in tomato ringspot virus (ToRSV, a nepovirus)-infected plants is associated with the AGO1-dependent translation repression of viral RNA2 [61]. Recently, SHIFTLESS (SFL), a broad-spectrum inhibitor of –1 ribosomal frameshifting, was isolated from Human T cell leukemia (MT4 cell) [98]. Ribosomal frameshifting is also a prevailing protein translation mechanism in plant viruses; thus, a similar ribosomal frameshifting inhibition mechanism might also exist in plants.

### 2.4. Atypical Dominant Viral Resistances

The exploration of viral resistance in the last decade also resulted in the discovery of some dominant resistance genes that are independent of the typical innate immunity signaling cassette for function. The products of these dominant resistance genes are structurally different from each other and from the typical R proteins, and, thus cannot be integrated into the conventional plant innate immunity. Based on the current understanding, most of these dominant resistance genes likely function by directly interacting with viral proteins to inhibit their activity. Hereafter, we referred to the products of these dominant resistance genes as ATYPICAL DOMINANT VIRAL RESISTANCE PROTEINs (ADVRPs).

The major ADVRPs discovered so far are members of the lectin protein family. Lectins, which are also known as agglutinins or jacalins, are a group of the carbohydrate-binding domain (CBD)-containing proteins that bind reversibly to specific monosaccharides or oligosaccharides [99]. Besides the CBD, many lectins also possess additional catalytic domains, such as glucanase, glycosidase, kinase, and RIP as well. Some kinase domain-containing lectins, such as DOES NOT RESPOND TO NUCLEOTIDES 1 (DORN1) and LIPOOLIGOSACCHARIDE-SPECIFIC REDUCED ELICITATION (LORE) from Arabidopsis and I-3 from tomato, function as the PRR receptors in PTI [100,101,102]. RESTRICTED TEV MOVEMENT (RTM) 1 is a lectin-like protein in Arabidopsis that specifically confers resistance to several potyviruses, e.g., TEV, lettuce mosaic virus (LMV), and plum pox virus (PPV), by restricting their long-distance movement [103,104]. The RTM1-mediated resistance is independent of HR, and neither involves SA nor induces SAR; instead, it is thought to form a complex in the phloem with at least two additional proteins, RTM2 (a small heat shock protein) and RTM3 (a meprin with the tumor necrosis factor receptor associated factor (TRAF) homology domain) and other host factors, as the products of the *RTM4* and *RTM5* genes, which were only genetically characterized [103,105,106,107,108,109]. The loss-of-function of either member in the complex results in the evanishment of the resistance, suggesting that RTM1-mediated resistance is dependent on the protein complex [108]. This protein complex likely functions by targeting the potyviral viral particle or CP-containing ribonucleoprotein complex in the phloem, as CP is involved in the RTM-mediated resistance breaking [110]. JACALIN-TYPE LECTIN REQUIRED FOR POTEXVIRUS RESISTANCE1 (JAX1) is another lectin-like ADVRP of Arabidopsis that confers broad-spectrum resistance to potexviruses at the early infection stage through recognizing viral RNA-dependent RNA polymerase (RdRp) and inhibiting its activity [111,112,113]. Other examples including a lectin-like protein (CIP-29) isolated from *Cyamopsis tetragonoloba* (L.) that is able to confer tobacco with resistance against the Sunn-hemp mosaic virus (SHMV, a tobamovirus) and a lectin from *Musa paradisiaca* (BanLec-1) that binds to the CP of TMV and prevents viral infection [114,115]. 

Other ADVRPs, such as the tomato Tm-1, which confers resistance to tomato mosaic virus (ToMV, a tobamovirus) [116] and VIOLAXANTHIN DEEPOXIDASE (ZmVDE) and h-type thioredoxin (ZmTrxh), which bestow maize with sugarcane mosaic virus (SCMV, a potyvirus) resistance [117,118], are structurally different from each other and cannot be classified into a typical protein family. Tm-1 is a T-cell immunoglobulin and mucin (TIM)-barrel-like domain-containing protein that is able to directly bind the replication proteins of ToMV and inhibit its activity [116]. ZmVDE functions by interacting with helper component-proteinase (HcPro), the VSR of SCMV, and attenuates its silencing suppression activity [118], whereas ZmTrxh inhibits viral RNA accumulation in the cytoplasm [117]. There are also several ADVRPs that probably do not function by interacting directly with viral proteins. For instance, STV11, a sulfotransferase from rice, confers rice with RSV resistance by catalyzing the conversion of SA into sulfonated SA [119]. Allelic *Ty-1* and *Ty-3* encode a γ-class RdRp that confers resistance to tomato yellow leaf curl virus (TYLCV, a begomovirus), and the hypermethylation of the genomic DNA has been hypothesized as the possible source of the resistance [120,121].

As research progresses, the number of ADVRPs will continue expanding. However, the origin of these ADVRPs is still a mystery at the present. A likely possibility is that these proteins have primary functions that are other than antiviral, and incidentally acquired the ability to inhibit virus proliferation [122]. For instance, Rsp5p, a yeast Nedd4 family E3 ubiquitin ligase that is normally involved in endocytosis [123], is able to interact with the replication proteins (p33 and p92) of tomato bushy stunt virus (TBSV, a tombusvirus), and mediates their degradation [124]. However, it is also possible that the resistance mediated by some ADVRPs is a conserved antiviral mechanism. For instance, lectins have been associated with antiviral activity since their discovery [99].

### 2.5. Ubiquitination and Autophagy-Mediated Protein Degradation

Ubiquitination-mediated protein degradation is the major protein turnover machinery in the plant. Ubiquitination is a multistep enzymatic reaction that is able to covalently attach ubiquitin (Ub) to the target proteins [125]. First, Ub precursor is proteolyzed by the Ub-activating enzymes (E1) and forms an E1–Ub intermediate in which the C-terminal glycine of Ub is linked via a thioester bond to a cysteine in E1. Next, the activated Ub is transferred to Ub-conjugating enzymes (E2). Finally, the Ub is delivered from the E2–Ub intermediate to target proteins with the help of Ub ligases (E3). Then, the ubiquitinated proteins are sent to the 26S proteasome for degradation. Ubiquitination-mediated protein degradation is pivotal for cellular protein homeostasis, and it is involved in nearly all cellular processes; therefore, it is not surprising that it is involved in almost all plant antiviral defense mechanisms. For instance, the full induction of *PR* genes’ expression in plant innate immunity involves the proteasome-mediated turnover of the transcription co-activator NPR1 [126]. Furthermore, the abundance of chitin receptor LYSIN MOTIF RECEPTOR KINASE5 (LYK5) is regulated by Arabidopsis E3 ubiquitin ligase PLANT U-BOX13 (PUB13) [127]. Unfolded protein response and endoplasmic reticulum (ER) stress, a complex and multifaceted intracellular signal pathway that is essential for reestablishing ER homeostasis, constitutes another example of plant response to viral infection regulated by ubiquitination-mediated protein degradation [128].

As the specificity of ubiquitination-mediated protein degradation is determined largely by E3, both plants and animals encode only a few E1s and E2s, and a large number of E3s. For instance, the model plant Arabidopsis encodes two E1s, 37 E2s, and more than 1400 E3s [129]. The masses of E3s allow plant cells to target, in addition to numerous endogenous proteins, exogenous proteins from invading pathogens. Hitherto, a group of E3 ligases was found to be able to specifically interact with typical viral proteins and mediate their degradation [130]. For instance, the tobacco really interesting new gene (RING) class E3 ligase *Nicotiana tabacum* RING FINGER PROTEIN 1 (NtRFP1) interacts with the βC1 protein encoded by the tomato yellow leaf curl China virus (TYLCCNV)-associated betasatellite, which is a viral pathogenesis factor, and VSR [131,132,133], to mediate its ubiquitination and degradation to attenuate virus proliferation [134]. Besides, the ubiquitination-mediated protein degradation is also involved in countering the RdRp of TYMV in Arabidopsis [135,136], and the CELL-DIVISION-CYCLE protein48 (CDC48), a conserved chaperone controlling protein fate in eukaryotes, extracts the movement protein (MP) of TMV from the ER to the cytosol for ubiquitination-mediated degradation [137,138]. However, whether the degradation of these viral proteins via ubiquitination is a defense mechanism of the plant or is the normal physiological homeostasis of the viral proteins needs to be carefully discriminated.

Autophagy is conserved machinery for transporting unwanted or misfolded protein aggerates or damaged organelles to the vacuole for degradation and recycling. In plants, it is pivotal for maintaining cellular homeostasis under normal conditions and resisting abiotic and biotic stresses [139]. Similar to ubiquitination, autophagy is also related to almost all aspects of cell physiological processes, including plant immunity [140,141]. Autophagy is involved in plant immunity in many ways, such as through balancing the homeostasis of immunity signaling components, degrading defense-related plant proteins, and regulating HR [140,142]. Plants also utilize selective autophagy to degrade virus-encoded protein aggregates, ribonucleoprotein, and even particles, which suggests that autophagy functions as a separate antiviral mechanism. Recently, several reports have highlighted the importance of this pathway in retarding virus proliferation: AUTOPHAGY-RELATED GENE6 (ATG6, also called BECLIN1), the core component of autophagy, interacts with NUCLEAR INCLUSION PROTEIN B (NIb), the RdRp of TuMV, to inhibit virus replication [143]. AUTOPHAGY-RELATED GENE8 (ATG8) specifically interacts with the βC1 of cotton leaf curl Multan virus (CLCuMuV)-associated betasatellite for degradation to impede its replication [144]. The autophagy cargo receptor, NEIGHBOR OF BRCA1 (NBR1), targets both unassembled CP and virus particles of CaMV to mediate their autophagy-dependent degradation, thereby restricting the establishment of CaMV infection [145]. Besides, NBR1 also suppresses TuMV accumulation by targeting HcPro, which is presumably in association with virus-induced RNA granules [146,147].

### 2.6. Cross-Talking between Different Antiviral Defenses

Plant antiviral mechanisms do not function separately; in contrast, there is vigorous cross-talking between different defense programs. In general, there are three types of cross-talking. Firstly, the component of one antiviral pathway is always regulated by other antiviral pathways. For instance, the transcripts of *R* genes are regulated by microRNAs, e.g., miR472, and further targeted by the *trans-acting si*RNAs (*ta-si*RNAs) or siRNAs produced from their transcripts by plant RNA-DEPENDENT RNA POLYMERASE 6 (RDR6), and the expression of the components of the RNA silencing pathway are also regulated by innate immunity responses [148,149,150]. Secondly, one mechanism needs the function of other pathways; for example, the HR of plant innate immune responses are regulated by autophagy and ER stress [140,151,152]. Finally, but most importantly, different types of mechanisms work cooperatively against plant viruses. For instance, the establishment of SAR involves SA-mediated innate immunity, RNA silencing, and even the protein secretory pathway [49,153]. SA is also involved in plant growth regulation by controlling gibberellin biosynthesis via endogenous small RNAs in the antiviral response to a potyvirus [154]. A calmodulin-like protein, REGULATOR OF GENE SILENCING CALMODULIN-LIKE PROTEIN (rgs-CaM), counterattacks intracellular VSRs by binding to their dsRNA-binding domains, mediates their degradation via the autophagy pathway, and also initiates SA-associated antiviral responses [155,156,157]. This sophisticated and dynamic cross-talk between different types of defense mechanisms allows the plant to respond quickly and efficiently to the infection of viruses and minimize the trade-off between defenses and normal development.

## 3. Strike Back from Viruses

Plant viruses have also gained multiple strategies to suppress and even exploit host defenses to ensure their successful infection [1]. Despite their small genome size (less than 20 kilobases) and that they only encode a few proteins, plant viruses have been endowed with great genome flexibility and multifunctional proteins to fight against the host’s sophisticated defense mechanisms. In general, plant viruses are able to strike back on every aspect of plant defense (Figure 1).

### 3.1. Mutation as a Master Mechanism for Escaping Host Antiviral Mechanisms

Due to the lack of proofreading viral RdRp, both plant and animal RNA viruses have a much higher mutation rate than that of their host cell, which is replicated via the DNA polymerase [158,159]. Plant DNA viruses, such as geminiviruses and nanoviruses, can evolve as quickly as their RNA counterparts [160,161]. As addressed in the previous section, plant antiviral mechanisms including PTI, ETI, and ADVRP-mediated resistance, are almost exclusively triggered by the recognition of a particular short sequence within the viral protein by PRRs, R proteins, or ADVRPs. As a result, it is easy for plant viruses to alter the amino acids responding to the perception to escape host immune response, which is a phenomenon called resistance breakdown. There are numerous such examples, for instance: one amino acid alteration in the VIRAL PROTEIN GENOME-LINKED (VPg) protein of *Rice yellow mottle virus* (RYMV, a sobemovirus) results in the loss of resistance of the *Oryza glaberrima* Tog7291 line carrying the *RYMV2* resistance gene [162]. The breakdown of sugar beet *Rz1*-mediated resistance against *Beet necrotic yellow vein virus* (BNYVV, a benyvirus) infection can be caused by a single mutation in its p25 coding sequence [163]. A single amino acid in the RdRp of PVX is responsible for JAX1-and Tm-1-mediated resistance breakdown [112,164]. Achieving long-term resistance through dominant resistance genes is a big challenge in crop breeding.

### 3.2. Inhibition of Innate Immunity

Despite the wealth of knowledge on how other phytopathogens and animal viruses suppress host immune responses [31,165], how plant viruses suppress plant innate immunity is just starting to be uncovered. Recently, several viral proteins, including the CP of PPV, the MP of CMV, and the P6 of CaMV, have been shown to be able to interfere with plant PTI signaling, including ROS production and SA accumulation, and eventually increase the susceptibility of the host to other pathogens [166,167,168]. However, the intracellular target of these viral proteins and how they suppress PTI is still elusive. More recently, we found that the NIb protein of TuMV can suppress host immune responses as well [169]. Moreover, we identified its role in the process of a host factor, SMALL UBIQUITIN-LIKE MODIFIER 3 (SUMO3), which can interact with and sumoylate NIb via a SUMO-interacting motif (SIM) at the C-terminal domain of the viral protein. Interestingly, suppression of host immune responses by NIb is dependent on SUMO3-mediated sumoylation [169]. SUMO3 is strongly and widely induced by SA and the defense elicitor Flg22 [170], and directly participates in plant innate immunity through fine-tune a regulating NPR1 function [171]. These results thus suggested that NIb functions through the disruption of the SUMO3 function to suppress plant antiviral immunity.

### 3.3. Suppression and Exploitation of Host RNA Silencing

Antiviral RNA silencing mainly occurs in the cytoplasm, where the replication of most plant RNA viruses takes place. To encounter host RNA silencing, the replication of many positive-sense single-strand RNA (+ssRNA) viruses are concealed in membranous inclusion bodies, vesicles, multivesicular bodies, or spherules that are remodeled from the plant cell endogenous membranes by viral proteins [172,173,174,175]. These membrane-sheltered virus replication factories allow the minimization of the disclosure of dsRNA that is generated by viral RdRps and is targeted by DCL proteins to trigger RNA silencing. All endogenous membranes, e.g., the ER, the membrane of chloroplasts, mitochondria, peroxisomes, and vacuole, have been exploited by plant +ssRNA viruses for replication. It is noteworthy that there is no obvious conservation in utilizing typical endomembranes among plant +ssRNA viruses of the same family or even of the same genus, and plant +ssRNA viruses from different families might use the same endomembrane to replicate. For instance, the carnation Italian ringspot virus (CIRV) and tomato bushy stunt virus (TBSV), which are both from the genus *Tombusvirus* within the family *Tombusviridae*, have very a similar genome structure, particle morphology, and replication strategy. However, these two viruses utilize the outer membrane of mitochondria and peroxisomes for replication, respectively [176,177]. Meanwhile, the TuMV (family *Potyviridae*), TMV (family *Virgaviridae*), and BMV (family *Bromoviridae*) hold replication vesicles or organelle-like structures derived from the ER [178,179,180]. Thereby, the endomembrane is selected “randomly” by the viruses to evade the evolutionary pressure imposed by host RNA silencing. The disturbance of plant endomembranes, especially the ER, by viruses always triggers ER stress responses, which are also implicated in plant reaction against other biotic and abiotic challenges and in normal development [181]. Interestingly, studies showed that the ER stress responses actually promote viral proliferation [182,183,184,185,186], suggesting that the ER stress signal pathway has been exploited by plant viruses for their own benefits.

Membranous virus replication factories are not sufficient to support robust viral replication, as abnormally abundant viral RNA in the plant cell can also trigger RNA silencing through the de novo synthesis of dsRNA by the host RdRp, which in turn inhibits viral protein synthesis and viral particle assembly. For instance, a CMV mutant that lacks the protein suppressing host RNA silencing (2b), can only replicate and cause systemic infection with extremely low efficiency [187,188]. Remarkably, most plant viruses, including RNA and DNA viruses, have evolved one or more VSRs to directly block the host RNA silencing mechanism [189]. A large number of VSRs encoded by varied plant viruses have been characterized. It is beyond the scope of this review to describe the mechanisms of action of each VSR in detail [189,190,191]. However, it is worth highlighting that the VSRs encoded by different viruses share no obvious amino acid sequence or structure similarity, suggesting they have independent origins. Nevertheless, VSRs can largely be divided into two classes based on their function mechanisms. The first class consists of VSRs functioning by sequestering viral dsRNA or vsiRNAs and/or by disrupting their biogenesis, such as the p19 of tombusviruses, NS3 of tenuiviruses, or the small replicase subunit (p122) of tobamoviruses. The second class consists of those functioning by subverting the components of the host RNA silencing pathway, such as the VPg of potyviruses, ßC1 of the betasatellite of begomoviruses, TGBp1 (p25) of potexviruses, and P0 of poleroviruses, although some VSRs, such as the HcPro of potyviruses and 2b of cucumoviruses, can both sequester vsiRNAs/dsRNA and disrupt RNA silencing components [190,191,192].

Plant viruses can also exploit the host RNA silencing system to promote their replication. Plant viruses exploit host RNA silencing mainly in two ways: one is by regulating endogenous miRNA expression, and another is by suppressing endogenous gene expression through viral-derived small-interfering RNAs (vsiRNAs). One such example is the infection of cymbidum ringspot virus (CymRSV, a tombusvirus), whose VSR (p19) can specifically upregulate the expression of miR168 to downregulate the transcripts of *AGO1*, which is a target of miR168 [193]. A similar phenomenon is involved in the breakdown of the RSV1-mediated resistance of the soybean cultivar PI96983 of the strain G7 of soybean mosaic virus (SMV, a potyvirus) [194]. Recently, Yang et al. showed that the intergenic siRNAs of TYLCV target a host long non-coding RNA to modulate disease symptoms [195]. Indeed, comparative analyses revealed that vsiRNAs have a wide-ranging and significant influence on the host transcriptome [196,197,198,199,200,201]. A more detailed outlook concerning how plant viruses utilize vsiRNAs to regulate endogenous gene expression can be found in these reviews [59,202].

### 3.4. Subversion of Translation Repression

A substantial amount of evidence suggests that animal viruses are able to subtly manipulate host protein synthesis machinery and adopt numerous unconventional translation mechanisms for a productive proliferation [203,204]. Plant viruses have also evolved numerous unconventional strategies that are both cap-dependent, and cap-independent to recruit the cellular translational machinery for the efficient synthesis of their proteins [205]. Thus, it is reasonable to believe that plant viruses are also able to “shut off” host protein biosynthesis, even though a direct link between the GCN1-GCN2-eIF2α regulatory pathway and plant virus infection has not been revealed yet.

NSP is the geminivirus-encoded membrane protein that counteracts NIK1-mediated translation repression [206]. NSP directly binds to the kinase domain of NIK1 and prevents its activation and the consequent downstream signaling by interrupting the phosphorylation of the key threonine residue at the position 474 [206]. The NSP–NIK interaction is conserved among geminivirus NSPs and NIK homologs from different hosts [206], suggesting that NIK-mediated translation suppression is a general plant antiviral defense that is successfully overcome by geminiviruses. Thus, NIK could be an excellent candidate for the development of broad-spectrum resistance against geminiviruses [207]. However, whether NIK-mediated translation repression is activated during RNA virus infection and has an antiviral effect against these viruses is still unknown at the present.

### 3.5. Prevention and Exploitation of Host Ubiquitination and Autophagy Pathways

Plant viruses also have developed several strategies to subvert host ubiquitination-mediated and autophagy-mediated antiviral machinery, e.g., by encoding deubiquitinases that target different endogenous and viral proteins [130]. For instances, the ubiquitination-mediated degradation of TYMV p66 can be counteracted by the interaction with viral 98K RdRp, which is also a functional deubiquitinating enzyme [208]. On the other hand, the βC1 protein disrupts the plant ubiquitination pathway by interacting with SPK1, and enhances geminivirus infection probably by subverting its own NtRFP1-mediated ubiquitination and degradation [209]. NBR1 suppresses TuMV infection by targeting HcPro protein for degradation [146]. However, NBR1-mediated HcPro degradation can be blocked by other viral proteins, such as VPg and 6K2 [146]. P6 protein, the major CaMV pathogenicity factor, suppresses SA-dependent autophagy [168], indicating that the autophagy-mediated restriction of CaMV proliferation could be diminished by P6.

Plant viruses are able to exploit ubiquitination and autophagy to promote their proliferation by encoding ubiquitin ligases or molecular adaptor proteins that target specific host factors for degradation. A large number of such examples have been reported, including the P0 protein of poleroviruses and enamoviruses, an F-box domain-containing protein, and PVX TGBp1 target AGO proteins for degradation via ubiquitin-mediated proteasome and autophagy pathways [210,211,212,213,214]; CLINK, another F-box domain-containing protein of faba bean necrotic yellows virus (FBNYV, a nanovirus), binds to SKP1 and the cell cycle protein pRB to promote virus replication [215,216]; rice stripe virus (RSV, a tenuivirus) MP (NSvc4) interferes with the S-acylation of remorin and induces its autophagic degradation to facilitate virus infection [217]; the geminivirus-encoded TRANSCRIPTIONAL ACTIVATOR PROTEIN (TrAP) protein, which is also known as C2, L2, AC2, or AL2, co-opts SKP1-CULLIN F-BOX PROTEIN (SCF)-mediated ubiquitination [218]; the VPg protein of potyviruses targets SUPPRESSOR OF GENE SILENCING 3 (SGS3) for degradation via both ubiquitination and autophagy pathways [219]. Recently, Yang et al. showed that the γb protein encoded by barley stripe mosaic virus (BSMV, a hordeivirus) can subvert host autophagy by disrupting the interaction between ATG7 and ATG8, which are two key regulators of the process [220], to promote viral infection [221]. Many animal viruses utilize autophagy machinery to assemble the membrane-bonded virus replication-associated structures [222,223,224,225,226]; however, a similar phenomenon has not been reported in plant +ssRNA viruses, even though the replication of most plant (+) RNA viruses are also associated with endomembranes [172].

## 4. Perspectives

Compared to the plant defense against other extracellular micropathogens, e.g., fungi and bacteria, in some aspects, plant antiviral defense could be more complex. This is apparent due to the intracellular parasitism of plant viruses, in which all genetic materials are directly explored by and connected with plant intracellular factors. Although this direct interaction allows the plant to evolve new defense mechanisms targeting viral factors, plant viruses can also take advantage of the interaction to explore the weak points of plant antiviral barriers and exploit their error-prone polymerases and multiple functional proteins to quickly escape host defenses and win the arms race. The ultimate goal of plant–virus interaction research is to establish sustainable virus resistance strategies to ensure food safety for the expanding human population. Indeed, considerable successes have been made in the management of viral diseases for varied crops in the last decade. For instance, *R* gene-directed breeding slowed down the ravages of many destructive plant viruses, such as tospoviruses, potyviruses, and begomoviruses [227,228]. Growing knowledge is also allowing researchers to develop engineered virus resistances, such as pathogen-derived resistance, RNA silencing-based resistance, pathogen-targeted resistance through zinc finger nucleases (ZiF), TRANSCRIPTION ACTIVATOR-LIKE EFFECTOR NUCLEASE (TALEN), and the CLUSTERED REGULARLY INTERSPACED PALINDROMIC REPEAT (CRISPR)/Cas9 system [229,230]. Small RNA-based genetic engineering has been commercially applied in engineering viral resistance for a number of crops, such as papaya, plum, squash, potato, pepper, and tomato [231]. Although multiple virus resistances have been achieved in several recent studies [232,233,234,235,236,237], preventing or retarding the virus resistance breakdown is still a major challenge in agriculture practice. Recently, several novel approaches have been proposed to this end, e.g., by stacking or pyramiding resistance genes, by altering known R proteins specificities and/or expanding resistance, by the exogenous application of dsRNA or siRNAs, and by the utilization of PRRs [238,239,240,241,242,243,244,245]. Nevertheless, how to develop novel antiviral strategies with broad-spectrum, efficient, sustaining, and environment-friendly resistances is still an open question.

Despite signs of progress in the last decade, many mysteries still need to be addressed, e.g., what molecular mechanisms are involved in incompatible interactions between plants and viruses, what virus-derived PAMPs and their respective PRRs exist, and what mechanisms underlie the mixed infection-associated resistance breakdown. For instance, the pre-infection of tomato cultivars that carry the *Sw-5* gene (which confers broad-spectrum resistance to tospoviruses through the recognition of a conserved 21-amino acid viral epitope [28]) with tomato chlorosis virus (ToCV, a crinivirus) results in susceptibility to tomato spotted wilt virus (TSWV, a tospovirus) [246]. Thus, these unexploited areas are especially worth investigating in the future.

## Figures and Tables

**Figure 1 viruses-11-00203-f001:**
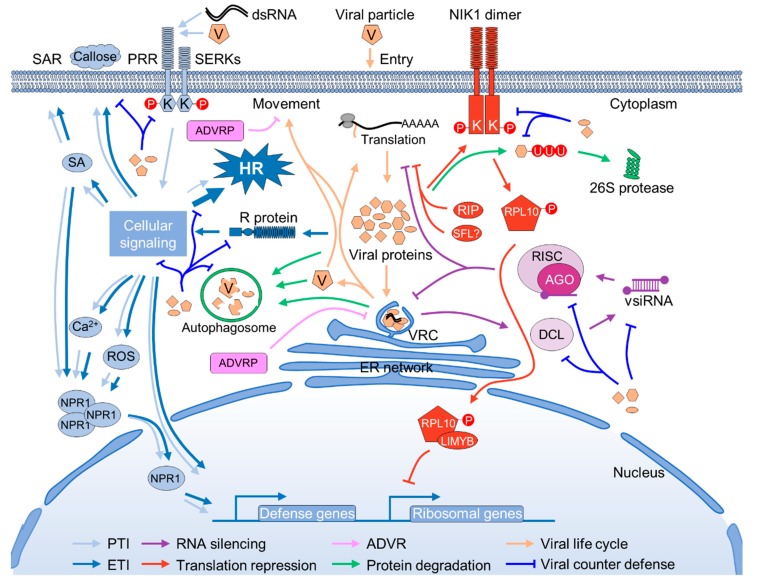
Schematic overview of plant antiviral pathways and viral counter-defenses. Pathogen-associated molecular patterns (PAMP)-triggered immunity (PTI), effector-triggered immunity (ETI), RNA silencing, ATYPICAL DOMINANT VIRAL RESISTANCE PROTEINs (ADVR), translation repression, and ubiquitination-mediated and autophagy-mediated protein degradation are indicated by light blue, dark blue, purple, red, pink and green arrows, or lines with bars, respectively. Virus infection circles are indicated by yellow arrows, and viral counter-defenses are indicated by blue lines with bars. Red circles with “P” and “U” letter indicate phosphorylation and ubiquitination, respectively. Unknown or putative paradigms are indicated as“?”.

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
