# Peer review of "The Tug-of-War between Plants and Viruses: Great Progress and Many Remaining Questions"

_viruses, 2019, doi:10.3390/v11030203_

Reviewer 1 Report

SUMMARY

Manuscript 438215 is a literature review focused on plant defense against viruses and mechanisms used by viruses to counter plant defense. The topic is interesting and the review is potentially informative. Additions and modifications are needed. Most of the sections referred to genes from publication and there is no critical assessment of their meaning as related to basic mechanisms of plant-virus interactions.

POINTS IN FAVOR

1. The topic is of potential interest to the area of plant-virus interactions.

2. The review points out areas of research with potential to improve our understanding of plant-virus interactions.

POINTS THAT NEED TO BE CLARIFIED 

1. Affiliation number 4 is not assigned to any of the authors.

2. Contrary to the statement in lines 26-28, plant viruses do not rely on the host cell for plant-to-plant transmission. This needs to be corrected.

3. Section 1.1. Innate immunity components, PAMP-triggered immunity and Effector-triggered immunity are presented mainly from the point of view of pathogens other than viruses. This reviewer strongly recommends that PAMP-triggered immunity and Effector-triggered immunity get presented based on current knowledge of viral pathogens.

4. Section 1.2. The interesting roles on RNA silencing in antiviral defense during compatible and compatible interactions need to be presented in detail.

5. Section 1.3. Two mechanisms of translational repression are presented: Global and specific.

These mechanisms need to be explained in detailed and related to viral infection. Lines 179 to 193 refer to a mechanism that has not been experimentally demonstrated. In contrast, siRNA-mediated translational dependent, which has been experimentally demonstrated, is mentioned and not explained.

6. Section 1.5. The role of ubiquitination and autophagy in plant-virus interactions need to be presented in an informative way. A critical assessment of the significance of the finding is needed as opposed to mentioning individual genes related to specific plant-virus combinations.

7. A figure is needed to illustrated/summarize/propose a model about our current understanding of the mechanisms of plant-virus interactions.

Author Response

Response to Reviewer 1 Comments

Point 1: Affiliation number 4 is not assigned to any of the authors.

Response 1: Thank you very much for pointing out this mistake, we have corrected this mistake accordingly.

Point 2: Contrary to the statement in lines 26-28, plant viruses do not rely on the host cell for plant-to-plant transmission. This needs to be corrected.

Response 2: Thank you very much for providing this valuable information, we have included it in the revision.

Point 3: Section 1.1. Innate immunity components, PAMP-triggered immunity and Effector-triggered immunity are presented mainly from the point of view of pathogens other than viruses. This reviewer strongly recommends that PAMP-triggered immunity and Effector-triggered immunity get presented based on current knowledge of viral pathogens.

Response 3: Thank you very much for this excellent suggestion. We have modified Section 1.1 to emphasis the current knowledge of PTI and ETI to viral pathogens.

Point 4 Section 1.2. The interesting roles on RNA silencing in antiviral defense during compatible and incompatible interactions need to be presented in detail.

Response 4: Thank you very much for this excellent suggestion. We modified the corresponding content to present important of RNA silencing in antiviral defence during compatible and incompatible plant-virus interactions.

Point 5: Section 1.3. Two mechanisms of translational repression are presented: Global and specific. These mechanisms need to be explained in detailed and related to viral infection. Lines 179 to 193 refer to a mechanism that has not been experimentally demonstrated. In contrast, siRNA-mediated translational dependent, which has been experimentally demonstrated, is mentioned and not explained.

Response 5: We totally agree with this excellent comment and, therefore, modified the content to emphasis there are two mechanisms of translational repression (global and specific). Besides, we have included the mechanism of small RNA-mediated translation repression in detailed in the revision.

Point 6: Section 1.5. The role of ubiquitination and autophagy in plant-virus interactions need to be presented in an informative way. A critical assessment of the significance of the finding is needed as opposed to mentioning individual genes related to specific plant-virus combinations.

Response 6: Thank you very much for this admirable suggestion. We have modified the content to point out the significance of ubiquitination and autophagy in plant-virus interaction.

Point 7: A figure is needed to illustrated/summarize/propose a model about our current understanding of the mechanisms of plant-virus interactions.

Response 7: Thank you very much for this brilliant suggestion. We totally agree with this comment, and a model has been included in the revision.

Reviewer 2 Report

This is a comprehensive and highly needed review about the complex interactions between plants and viruses. The review is very well organized and seem to address all major aspects related to its title. However, it lacks figures to illustrate the different pathways mentioned and the text requires extensive editing for correct English language and style.  

Comments:

- Generally, the text requires editing, particularly in its first half. Several sentences require complete rephrasing.

- include figures to illustrate the interactions and pathways.

- the title of the manuscript does not sound correct to me. I suggest changing it to "The tug-of-war between plants and viruses: lots of progress and many remaining questions"

- line 26: “obligate noncellular parasites”; you mean “intracellular” parasites? Moreover, why are virus considered as “parasites”? Surveys indicate that most plant:virus interactions do not to cause any symptoms. Given that viruses are drivers of evolution and given that they live in tight interactions with the cell should they not rather be considered as “obligate symbionts”?

- line 30: I do not like the expression “ the interaction network can be classified into two aspects”; I would suggest something like “the interactions network includes mechanisms by which the plant…..and by which the virus……”).

- line 33: clarify: “confrontation” between what?

- line 33: it is correct that the balance between the mechanisms used by the plant and the virus determine the outcome of the interaction. However, it is problematic to describe resistance as an incompatible interaction. For example, resistance mechanisms like RNA silencing and PTI cause antiviral resistance but nevertheless are part of compatible interactions.  Incompatible interactions are an extreme case usually based on recessive or dominant resistant genes which either prevent infection (e.g. lack of a factor required for infection; recessive resistance) or cause the death of infected cells (dominant resistance gene).

It could be mentioned that the balance between the mechanisms used by the virus and the host may determine whether a virus is pathogenic or not. For example, if the virus is not recognized by the plant defense mechanism (e.g. by PTI) it will likely be pathogenic. Pathogenicity also likely occurs if a virus expresses too much VSR activity. This has been nicely illustrated by Korner et al (citation 61).

Line 52: not all viruses are “pathogenic”

Line 73: “viewed as non-PAMP or effector coding pathogens”. Unclear sentence. The term “effector” has not been used and explain up to this point. Moreover, again, viruses are not pathogens per se.

Line 95: The statement that CaLecRK-S.5 could be a PRR or its cofactor appears highly speculative. Based on the evidence mentioned, this factor could play any role in defense signaling or activity.

Line 100: delete “very”.

Lines 114-118: In this context it could be worthwhile to mention that the N gene is temperature sensitive and allows further infection as soon as it is inactivated at higher temperature. See, for example Wright et al., 2000 (PMID: 10938355).

Lines 165 -177: This text is hard to read. In line 165, before “1)”, the text should say something like “similar to innate immunity, RNA silencing also….. Similarly, in line 170, before “1)”, the text should say something like “Unlike PTI and ETI, RNA silencing….

Line 166: What do you mean with “extraordinary dsRNA molecules”?

Line 166: replace “base-independent” by “nucleotide sequence-independent”

Line 167: use “cell-to-cell and systemic movement” instead of “motion”?

Lines 176-177: I do not understand why it is here proposed to consider RNA silencing and PTI as sperate plant antiviral mechanisms. It is clear that these two mechanisms are different. For example PTI depends on PR gene expression and is protein mediated whereas RNA silencing is RNA-mediated.

Line 203: “There was a blossom in seeking to utilize”….?

Line 306: A study showed that TMV MP degradation depends on its extraction from the ER by CDC48, which could be mentioned here (PMID: 23027663)

Line 322: It could be mentioned in this context that NBR1 also targets HC-Pro of TuMV. This phenomenon is only mentioned later in the text (citation 196).

Line 329: R genes are rather regulated by production of trans-acting siRNAs from their transcripts rather being targeted by such siRNAs.  In this context, the publication by Boccara et al (PMID: 24453975) could be mentioned. These authors showed that rdr6 mutants exhibit constitutive PTI thus indicating that RDR6 induced during RNA silencing downregulates PTI resistance genes.  

Line 416: Given that TMV is mentioned several times in the article, I suggest to also mention the TMV VSR (the small replicase subunit), shown to bind sRNA, to interfere with sRNA methylation, and to inhibit RISC assembly (Csorba et al., 2007; Vogler et al., 2007).

Paragraph about perspectives: The perspectives could include the exogenous treatment of plants with dsRNA or siRNAs for antiviral crop protection (e.g. PMID:29479789; PMID:28067898).

Author Response

Response to Reviewer 1 Comments

This is a comprehensive and highly needed review about the complex interactions between plants and viruses. The review is very well organized and seem to address all major aspects related to its title. However, it lacks figures to illustrate the different pathways mentioned and the text requires extensive editing for correct English language and style.

Point 1: Generally, the text requires editing, particularly in its first half. Several sentences require complete rephrasing.

Response 1: Thank you very much for this admirable suggestion. We have asked an English native to correct the manuscript and then applied the MDPI English Editing service to further correct the English.

Point 2: Include figures to illustrate the interactions and pathways.

Response 2: Thank you very much for these excellent suggestions. We totally agree with this comment, and a figure has been included in the revision to illustrate the interactions and pathways.

Point 3: the title of the manuscript does not sound correct to me. I suggest changing it to "The tug-of-war between plants and viruses: lots of progress and many remaining questions"

Response 3: Thank you very much for this excellent suggestion. We have corrected the title accordingly.

Point 4 line 26: “obligate noncellular parasites”; you mean “intracellular” parasites? Moreover, why are virus considered as “parasites”? Surveys indicate that most plant:virus interactions do not to cause any symptoms. Given that viruses are drivers of evolution and given that they live in tight interactions with the cell should they not rather be considered as “obligate symbionts”?

Response 4: Thank you very much for this nice comment. We have corrected this mistake accordingly.

Point 5: line 30: I do not like the expression “ the interaction network can be classified into two aspects”; I would suggest something like “the interactions network includes mechanisms by which the plant…..and by which the virus……”).

Response 5: Thank you very much for this comment. We have corrected these sentences accordingly.

Point 6:  line 33: clarify: “confrontation” between what?

Response 6: Thank you very much for this comment. We have re-phrased the corresponding sentence.

Point 7:  line 33: it is correct that the balance between the mechanisms used by the plant and the virus determine the outcome of the interaction. However, it is problematic to describe resistance as an incompatible interaction. For example, resistance mechanisms like RNA silencing and PTI cause antiviral resistance but nevertheless are part of compatible interactions.  Incompatible interactions are an extreme case usually based on recessive or dominant resistant genes which either prevent infection (e.g. lack of a factor required for infection; recessive resistance) or cause the death of infected cells (dominant resistance gene).

Response 7: Thank you very much for this superb comment. We totally agree with this comment, we therefore integrated this idea into the revision.

Point 8: It could be mentioned that the balance between the mechanisms used by the virus and the host may determine whether a virus is pathogenic or not. For example, if the virus is not recognized by the plant defense mechanism (e.g. by PTI) it will likely be pathogenic. Pathogenicity also likely occurs if a virus expresses too much VSR activity. This has been nicely illustrated by Korner et al (citation 61).

Line 52: not all viruses are “pathogenic”

Response 8: Thank you very much for this excellent comment. We totally agree with this comment, we therefore integrated this idea into the revision.

Point 9: Line 73: “viewed as non-PAMP or effector coding pathogens”. Unclear sentence. The term “effector” has not been used and explain up to this point. Moreover, again, viruses are not pathogens per se.

Response 9: Thank you very much for this comment. We corrected the sentence accordingly.

Point 10: Line 95: The statement that CaLecRK-S.5 could be a PRR or its cofactor appears highly speculative. Based on the evidence mentioned, this factor could play any role in defense signaling or activity.

Response 10: Thank you very much for this comment. We have re-phrased corresponding content to tone-down the conclusion.

Point 11: Line 100: delete “very”. Lines 114-118: In this context it could be worthwhile to mention that the N gene is temperature sensitive and allows further infection as soon as it is inactivated at higher temperature. See, for example Wright et al., 2000 (PMID: 10938355).

Response 11: Thank you very much for this brilliant comment. We have included this information and the citation in the revision.

Point 12:  Lines 165 -177: This text is hard to read. In line 165, before “1)”, the text should say something like “similar to innate immunity, RNA silencing also….. Similarly, in line 170, before “1)”, the text should say something like “Unlike PTI and ETI, RNA silencing….

Response 12: Thank you very much for this nice comment. We have rephrased the corresponding sentences accordingly.

Point 13:  Line 166: What do you mean with “extraordinary dsRNA molecules”? Line 166: replace “base-independent” by “nucleotide sequence-independent” Line 167: use “cell-to-cell and systemic movement” instead of “motion”?

Response 13: Thank you very much for this comment. We have corrected accordingly.

Point 14: Lines 176-177: I do not understand why it is here proposed to consider RNA silencing and PTI as sperate plant antiviral mechanisms. It is clear that these two mechanisms are different. For example PTI depends on PR gene expression and is protein mediated whereas RNA silencing is RNA-mediated.

Response 14: Thank you very much for this comment. We have corrected accordingly.

Point 15: Line 203: “There was a blossom in seeking to utilize”….?

Response 15: Thank you very much for point out this mistake. We have rephrased the sentence accordingly.

Point 16: Line 306: A study showed that TMV MP degradation depends on its extraction from the ER by CDC48, which could be mentioned here (PMID: 23027663). Line 322: It could be mentioned in this context that NBR1 also targets HC-Pro of TuMV. This phenomenon is only mentioned later in the text (citation 196).

Response 16: Thank you very much for these excellent comments. We have included these information and citations in the revision.

Point 17: Line 329: R genes are rather regulated by production of trans-acting siRNAs from their transcripts rather being targeted by such siRNAs.  In this context, the publication by Boccara et al (PMID: 24453975) could be mentioned. These authors showed that rdr6 mutants exhibit constitutive PTI thus indicating that RDR6 induced during RNA silencing downregulates PTI resistance genes. 

Response 17: Thank you very much for this comment. We totally agree with this comment and have included this information in the revision.

Point 18: Line 416: Given that TMV is mentioned several times in the article, I suggest to also mention the TMV VSR (the small replicase subunit), shown to bind sRNA, to interfere with sRNA methylation, and to inhibit RISC assembly (Csorba et al., 2007; Vogler et al., 2007).

Response 18: Thank you very much for this nice comment. We have included TMV VSR in the revision.

Point 19: Paragraph about perspectives: The perspectives could include the exogenous treatment of plants with dsRNA or siRNAs for antiviral crop protection (e.g. PMID:29479789; PMID:28067898).

Response 19: Thank you very much for this admirable comment. We have included this suggestion in the revision.

Reviewer 3 Report

This paper is a review article titled by “The tug-of-war between plant and viruses: lots of progresses and many questions remained”. The title and theme seem timely and attractive at this time point and contains recent information in this field citing papers in up to 2018. Apparently this manuscript covers a very wide range of issues citing over 230 references. It takes time for readers to grasp the points of this complex and rapidly growing field at a glance. From this point of view it is strongly recommended to add some appropriate/concise Table or Figure in the text for showing a list of defense/offense categories from both host and virus sides containing list of isolated genes with isolated genes including CaLecRK-S.5, LAX1, N, Tm-1, Tm-2, Tm-22, RCY1 BV1, Rsv1 L1, L2, L3, P3 etc (all known genes so far). This kind of Table or Figure could be much more helpful for readers to understand. There are some errors and mistakes. Also language improvements are needed in the text. Some comments are described below.

Comments:

Line 91: Pepper mild mottle virus is a tobamovirus instead of a potyvirus. PMMV should be PMMoV.

Line 126: Explain “HRT”. This is used only here.

Line 167: “motion” may be replaced by other term such as movement, transmission or transfer ?

Lines 179-193: Although this review excludes recessive genes, it would be better to put one sentence describing the importance of translation step, which is deeply related with virus resistance as mutated genes (recessive genes). 

Line 192: “this role has been completed quelled” needs to be corrected.

Line 224: Any appropriate reference is cited here.

Lines 795 and 796: Correct capital letters or lower case letters as the guideline shows

Line 1051: 2017 should be in bold.

Author Response

Response to Reviewer 1 Comments

Point 1: This paper is a review article titled by “The tug-of-war between plant and viruses: lots of progresses and many questions remained”. The title and theme seem timely and attractive at this time point and contains recent information in this field citing papers in up to 2018. Apparently this manuscript covers a very wide range of issues citing over 230 references. It takes time for readers to grasp the points of this complex and rapidly growing field at a glance. From this point of view it is strongly recommended to add some appropriate/concise Table or Figure in the text for showing a list of defense/offense categories from both host and virus sides containing list of isolated genes with isolated genes including CaLecRK-S.5, LAX1, N, Tm-1, Tm-2, Tm-22, RCY1 BV1, Rsv1 L1, L2, L3, P3 etc (all known genes so far). This kind of Table or Figure could be much more helpful for readers to understand. There are some errors and mistakes. Also language improvements are needed in the text. Some comments are described below.

Response 1: Thank you very much for these excellent comments. We have included and Figure to allow readers to get a glance of this article and send the manuscript to MDPI language service for language improvement.

Point 2: Line 91: Pepper mild mottle virus is a tobamovirus instead of a potyvirus. PMMV should be PMMoV.

Response 2: Thank you very much for point out this mistake. We have corrected accordingly.

Point 3: Line 126: Explain “HRT”. This is used only here.

Response 1: Thank you very much for this nice suggestion. We have added the full name of HRT in the revision.

Point 4: Line 167: “motion” may be replaced by other term such as movement, transmission or transfer ?

Response 1: Thank you very much for this admirable suggestion. We have corrected accordingly.

Point 5: Lines 179-193: Although this review excludes recessive genes, it would be better to put one sentence describing the importance of translation step, which is deeply related with virus resistance as mutated genes (recessive genes).

Response 1: Thank you very much for this excellent suggestion. We have added a sentence describing the important of translation and its relationship with recessive genes.

Point 6: Line 192: “this role has been completed quelled” needs to be corrected.

Response 1: Thank you very much for this nice comment. We have corrected accordingly.

Point 7: Line 224: Any appropriate reference is cited here.

Response 1: Thank you very much for this nice comment. A relevant reference has been added in the revision.

Point 8: Lines 795 and 796: Correct capital letters or lower case letters as the guideline shows

Response 1: Thank you very much for point out this mistake. We have corrected accordingly.

Point 9: Line 1051: 2017 should be in bold.

Response 1: Thank you very much for point out this mistake. We have corrected accordingly.

 Round  2

Reviewer 1 Report

The authors responded to my comments on the initial revision and included a figure summarizing a model.